# Enhancing Lung Cancer Histopathological Subtyping *via* Fuzzy Patch Scoring Integrated into an Ensemble Deep Learning Framework

Mohammad Mehdi Hosseini
Department of Pathology
SUNY Upstate Medical University
Syracuse, NY 13210
hosseini_mm@yahoo.com

Bardia Rodd
Department of Pathology
SUNY Upstate Medical University
Syracuse, NY 13210
roddb@upstate.edu

*Abstract*—Automated histopathological subtyping of lung cancer from stained whole-slide images (WSIs) remains a pivotal yet challenging task due to pronounced tumor heterogeneity, complex cellular morphology, and severe class imbalance in existing datasets. Deep learning models vary in their ability to capture pathomic diversity, and their diagnostic performance is closely tied to the quality of tissue patches extracted from WSIs. To address these challenges, we propose a novel ensemble deep learning framework augmented with fuzzy-weighted patch quality assessment to optimize the selection and weighting of informative regions. High-quality patches are identified using a fuzzy scoring mechanism and processed through multiple pre-trained convolutional neural networks (CNNs) and vision transformer (ViT) models to extract diverse feature representations. These are integrated via latent embeddings, with fuzzy scores incorporated both as auxiliary inputs and as weights in the loss function, reinforcing attention to clinically relevant regions. Our method outperformed current state-of-the-art models by 1.5% and 1.4% (CI: 95%), achieving accuracies of 96.1% on BMIRDS-LUAD and 93.0% on WSSS4LUAD, demonstrating enhanced robustness in subtype classification and strong potential for clinical integration.

*Index Terms*—Digital pathology, Latent space embedding, Ensemble model, Fuzzy scoring, Transfer learning.

## I. INTRODUCTION

Recent advancements in high-throughput slide scanning have profoundly advanced digital pathology, facilitating high-resolution analysis of whole-slide images (WSIs) [1]. These technological strides enable unprecedented detail in histopathological examination, yet the challenge of achieving effective patch-level representation persists due to the gigapixel scale of WSIs, marked tumor heterogeneity, and intricate cellular morphologies. Extracting high-quality patches is paramount for training robust deep neural networks, but pinpointing diagnostically relevant regions within these expansive image spaces remains a formidable obstacle. Contemporary patch selection methodologies frequently grapple with significant limitations, including suboptimal clustering techniques [2], heightened vulnerability to overfitting [1], and pronounced class imbalances within datasets [2]. Compounding these issues are additional barriers, such as the paucity of expert annotations [3], variability in tissue staining protocols [3], substantial computational requirements [4], and the persistent inclusion of redundant or non-informative patches [4]. These challenges collectively impede the development of classification models that are both precise and generalizable across diverse clinical contexts.

To surmount these obstacles, we introduce a sophisticated fuzzy-weighted patch quality framework integrated within an ensemble deep learning architecture. This innovative framework employs a fuzzy scoring strategy to prioritize patches that are highly representative of specific subtypes, thereby enhancing the quality and relevance of the training dataset. By leveraging multiple pre-trained networks, including convolutional neural networks (CNNs) and transformers, our approach extracts a diverse array of pathomic features, which are subsequently harmonized through multimodal embeddings to achieve a comprehensive and cohesive representation. Our methodology delivers four pivotal contributions:

1) A novel fuzzy scoring mechanism meticulously designed to identify and select diagnostically salient patches, ensuring that only the most informative regions guide model training;

2) An ensemble learning strategy that capitalizes on heterogeneous deep architectures to enhance robustness and diversity in feature extraction, mitigating the limitations of single-model approaches;

3) A unified latent embedding model that aligns disparate feature representations prior to integration, fostering seamless fusion and improved predictive accuracy;

4) A rigorous comparative evaluation against state-of-the-art frameworks, providing a robust benchmark to validate the superior performance and generalizability of our approach.

Contemporary methods still face limitations such as poor generalization due to patch redundancy [5], overfitting from indiscriminate patch inclusion [5], sensitivity to small/distant objects [6], and inconsistent evaluation metrics [5], [6].

Our proposed framework is the first to integrate fuzzy logic with ensemble learning for subtype classification in whole-slide images of invasive nonmucinous adenocarcinoma. Our gap-bridging fuzzy-guided framework evaluates subtype-specific coverage at the patch level and leverages this score for both feature fusion and loss computation. Empirical results consistently demonstrate performance gains over state-of-the-art methods, underscoring the significance of adaptive, quality-driven patch selection in advancing AI-enabled histopathological diagnostics.

## II. RELATED WORK

Many recent studies in histopathology [5]–[7], [22] have explored quality-based patch selection. However, even with such systems, challenges persist, including the learning of non-representative features, elevated risk of overfitting, and the need for complex analytical pipelines. Additionally, patch detection methods often struggle with identifying small or spatially distant pathological structures, which adversely impacts patch quality within model training.

Fuzzy logic-based systems have demonstrated potential in interpreting pathology laboratory reports, managing uncertainty in medical data [8], and addressing fuzzy boundaries in clinical image segmentation models [8]; however, their application to quality patch selection remains unexplored. Ensemble deep learning strategies have also shown notable success [7], [8], yet existing approaches do not address the integration and alignment of latent feature representations across these models. To the best of our knowledge, our study is the first to introduce a latent space feature embedding strategy for enhancing patch-based classification performance.

## III. METHOD

### A. Notations and Preliminaries

For an input WSI, $WS_i \in \mathbb{R}^{d_x \times d_y \times 3}$, representing the $i$-th patient's slide, we tessellate it into small patches, extract embeddings using a pre-trained vision encoder, and aggregate them into a slide-level representation. Tissue regions are detected and segmented to exclude background using the HistoQC toolbox [9] and a tissue detector trained on mask annotations, yielding $H_i \subseteq WS_i, H_i \in \mathbb{R}^{t_x \times t_y \times 3}, t_x \leq d_x, t_y \leq d_y$.

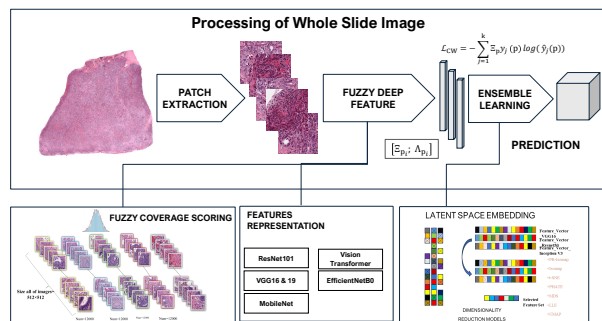

Fig. 1: Whole Fuzzy Coverage Deep Ensemble Learning Classifier Framework.

Non-overlapping patches are then extracted, with fuzzy scoring applied to ensure high-quality patch selection in pathology images. Our method processes an input pathology image, $H_i$, by tessellating it into non-overlapping patches, $p \in P = \{p_1, p_2, \ldots, p_n\}$, such that:

$$H_i = \sum_{i=1}^{n} p_i, \quad p_i \in \mathbb{R}^{a \times b} \tag{1}$$

where patches are defined with dimensions $a, b = 512$, and the total number of patches, $n$, is dynamically determined based on image resolution. These patches are grouped into distinct sets $s_j, j \in \{1, 2, \ldots, k\}$, corresponding to cancer subtypes, where $k = 5$ represents lung subtypes: acinar, lepidic, micropapillary, papillary, and solid. The set of patches for subtype $j$ is denoted as:

$$P_{s_j} = \bigcup_{H_i \subset s_j} p_i, \quad \neg \forall s_j = s_m, j \neq m \tag{2}$$

with a total patch count of $N_{s_j}$, where subset balance is not guaranteed ($\neg \forall s_j = s_m, j \neq m$). To address this imbalance, we propose a balanced sampling strategy. For each subtype $s_j$, we randomly select $\tilde{N}_{s_j}$ patches from the overall dataset, forming a balanced subset $\tilde{P}_{s_j} \subseteq P_{s_j}$. The final balanced dataset is represented as:

$$\tilde{P}_s = \bigcup_{j=1}^{k} \tilde{P}_{s_j} \tag{3}$$

This ensures equitable patch representation across subtypes during model training, reducing bias toward majority classes.

### B. Fuzzy Labeling for Tissue Coverage

We propose a novel fuzzy labeling mechanism to address the semantic ambiguity inherent in complex histopathology patches. Traditional hard-labeling approaches struggle to capture nuanced tissue patterns in high-resolution WSIs, where heterogeneous tissue

compositions often coexist within a single patch. Our fuzzy labeling framework quantifies subtype representation by assessing the proportional coverage of cancerous regions within each patch, providing a more granular and biologically relevant annotation strategy. For each patch in the balanced dataset, the pre-segmented subtype region is denoted as $m_s(p_i)$, formally defined as:

$$m_s(p_i) = \{\forall p_i \in \tilde{P}_s \mid p_i \subseteq s_j\}, j = \{1, \ldots, k\} \quad (4)$$

Here, we define fuzzy scoring functions, $\Xi(p_i)$ and $\Lambda(p_i)$, to quantify the representation of targeted subtypes within the designated patch regions, formulated as follows:

*1) Definition 1.: The Coverage Value, $\Xi(p_i)$, for a given patch $p_i$ is defined as the proportion of the designated subtype region relative to the total patch area:*

$$\Xi(p_i) = \frac{\sum_{p_i} m_{s_j}(p_i)}{M_{p_i}}, \quad \Xi(p_i) \in [0, 1] \quad (5)$$

*where $M_{p_i}$ defines as:*

$$M_{p_i} = \sum_{p_i} m_s(p_i) + \sum_{p_i} m_{\neg s}(p_i).$$

The patch coordinates are $a$ and $b$. This metric ensures a precise scoring mechanism for the targeted subtypes.

*2) Definition 2: The Outbound Value, $\Lambda(p_i)$, is defined as the complementary measure to the Coverage Value $\Xi(p_i)$:*

$$\Lambda(p_i) = 1 - \Xi(p_i) = \frac{\sum_{p_i} m_{\neg s_j}(p_i)}{M_{p_i}}, \quad \Lambda(p_i) \in [0, 1]. \quad (6)$$

The fuzzy scores $\Xi(p_i)$ and $\Lambda(p_i)$ provide a continuous representation of tissue composition within a patch, ranging from 0 for complete absence to 1 for full occupancy by the designated subtype. This refined labeling strategy effectively captures the gradual morphological transitions within histopathological samples. These values are subsequently stored alongside the patch data for seamless integration into the deep learning framework. Notice that patches not related to tumor subtype annotations, such as benign or background, were removed from the dataset. $\tilde{P}_s$. This ensured that all fuzzy-scored patches were classified as malignant subtypes.

### C. Fuzzy Ensemble Learning

We adopt several pre-trained deep learning architectures to obtain a comprehensive feature representation, enhancing classification robustness. Let $\Phi_{\theta_e} : \mathbb{R}^q \to \mathbb{R}^d$ represent a pretrained deep neural network with parameters $\theta_e$, where $e \in \{1, \ldots, \omega\}$, trained on an independently and identically distributed (i.i.d.) dataset sampled from a distribution D. Each input $x \subseteq \tilde{P}_{s_j}$ provides a representation of the patch in the feature space,

$\mathcal{F}_{\theta_e} \in \mathbb{R}^d$. The model consists of a feature extractor $h_{\psi_e} : x \to \mathbb{R}^q$ and a linear classifier characterized by parameters $\{w_k, b_k\}$, such that $\Phi_{\theta_e}(x) = w_k^\top h_{\psi_e}(x) + b$, where $\mathcal{F}_{\theta_e} = \Phi_{\theta_e}(x; \theta_e)$ denotes the feature extraction function of model e with parameters $\theta_e$, applied to a random patch x.

*1) Latent Representation:* We project an ensemble of features acquired from the pretrained models to a lower-dimensional space using various embedding functions, $g_i$ [10]–[13], $g_i : \mathbb{R}^d \to \mathbb{R}^c$, $c \ll d$. This projection is defined as $G_i : \mathcal{F}_{\theta_e} \to \bar{\mathcal{F}}_{\theta_e}$.

*2) Fuzzy Classifier Module:* We incorporate fuzzy information obtained from the functions defined in (3.2) as a feature enhancement strategy to improve the model's ability to discern relevant patterns. A dense layer $\Psi_{\bar{\mathcal{F}}}(v; \theta_{\bar{\mathcal{F}}})$ with limited neurons, parameterized by $\theta_{\bar{\mathcal{F}}}$ and utilizing ReLU activation, processes the two-dimensional vector, $v = [\Xi_{p_i}; \Lambda_{p_i}]$. This augments the feature representation using coverage and outbound values of $\bar{\mathcal{F}}_\theta$. The enhanced features are then used to classify cancer subtypes through a Multi-Layer Perceptron (MLP) deep learning architecture.

*3) Coverage-Weighted Loss Function:* To further enhance diagnostic accuracy, we modify the internal loss function by incorporating the coverage value $\Xi_{p_i}$ into the Categorical Cross-Entropy loss function, resulting in the Coverage-Weighted Loss Function $L_{CW}$, defined as:

$$L_{CW} = -\sum_{j=1}^{k} \Xi_p y_j(p) \log \hat{y}_j(p), \quad (7)$$

where $\hat{y}(p)$ represents the predicted probability distribution. The ensemble neural network is trained with this modified loss function, performing backpropagation of gradients weighted by the coverage value $\Xi_p$.

## IV. Experiments and Results

### A. Datasets and Evaluation Metrics

*1) BMIRDS Lung Data.:* To evaluate our proposed models, we analyzed 203,226 histopathology patches extracted from 143 H&E-stained, formalin-fixed paraffin-embedded (FFPE) WSIs of non-mucinous adenocarcinoma, comprising five subtypes: lepidic, acinar, papillary, micropapillary, and solid, sourced from the DHMC repository. All WSIs were labeled based on the consensus of three pathologists [14].

*2) GDPH-LUAD and TCGA-LUAD data.:* 67 H&E-stained slides from GDPH-LUAD and 20 WSIs from TCGA-LUAD, obtained from the WSSS4LUAD dataset [15], contributed 10,091 patches to the training set, with a label distribution of tumor: 6,579, stroma: 7,076, and normal: 1,832, with their masks.

## B. Implementation Details

The initial high-dimensional feature vector, $\mathcal{F}_{\theta_e} \in \mathbb{R}^d$, is reduced to $\bar{\mathcal{F}}_{\theta_e} \in \mathbb{R}^c, c = 100$ via embedding. This feature representation is further refined through deep learning enhancement as $\bar{\bar{\mathcal{F}}}_{\theta_e}$ before serving as input to our classifier.

The dataset $\tilde{P}_s$ is split into training (70%), validation (15%), and test (15%) sets. Model training utilizes the Adam optimizer with a learning rate of $1e^{-4}$, $\beta_1 = 0.9$, $\beta_2 = 0.999$, and early stopping with a patience of 10 epochs based on validation accuracy. Performance is assessed using Accuracy, Precision, Recall, and F1-score. We used ground-truth to calculate fuzzy scores. The deep learning model was implemented in Python 3.9 using TensorFlow 2.8 with the Keras API. We tested our approach with multiple pre-trained models, including ResNet-50, ResNet-101 [16], VGG16, VGG19, InceptionV3 [17], MobileNet, EfficientNetB0, and Vision Transformer [18], all initialized with 'IMAGENET' weights. Models were trained with a batch size of 32 for up to 100 epochs, with early stopping based on validation accuracy.

## C. Feature Extraction and Fusion Results

We assess the efficacy of our multi-model feature fusion strategy by comparing the classification performance of models using individual pre-trained CNNs against the fused feature vector $\mathcal{F}_{\theta_e}$. This initial evaluation inputs extracted features directly into the ensemble neural network classifier without feature reduction or fuzzy labels. Test set classification performances for individual models and the fused feature vector are presented in Table I.

The fused feature vector $\mathcal{F}_{\theta_e}$ consistently outperforms individual CNN feature sets across all metrics, achieving an accuracy of 86.2%, an F1-score of 86.0%, and an AUC of 97.0%. These results underscore the advantage of combining representations from diverse deep learning architectures to capture complementary information.

As presented in Table I, the fused feature vector $\mathcal{F}(p)$ outperforms individual CNN feature sets across all metrics, achieving an accuracy of 86.2%, an F1-score of 86.0%, and an AUC of 97.0%. These results demonstrate the power of integrating diverse deep learning architectures to capture complementary information.

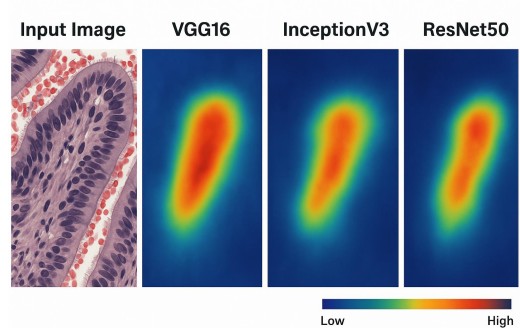

Fig. 2: Attention map obtained for VGG16, InceptionV3, and ResNet50.

## D. Impact of Ensemble Learning

We assessed our Ensemble Neural Network ($\varepsilon$) against a baseline single-layer Dense classifier, both trained on the PCA-reduced fused feature representation $\bar{\bar{\mathcal{F}}}_{\theta_e}$ ($d_{\text{reduced}} = 200$, determined via Gap statistics) under identical experimental conditions. As shown in Table II, the ensemble model achieved a 2.6 percentage point increase in accuracy and outperformed the baseline across all key metrics, highlighting its ability to leverage complex, high-dimensional features for superior classification.

Class activation heatmaps (Fig. 2) confirm that the backbone networks (VGG16, InceptionV3, and ResNet50) focus on tumor cell nuclei, though with varying contours, shapes, and intensities. These differences support our hypothesis that ensemble methods, by integrating diverse features, capture more comprehensive information, leading to improved performance, as evidenced in Table III.

## E. Ablation analyses

To ensure a robust evaluation, we use overall accuracy (ACC) and precision as primary metrics to assess model performance. We conducted an ablation analysis to evaluate the effectiveness of our proposed fuzzy scoring block and to determine the contribution of individual model components to classification performance. This analysis quantifies the impact of the fuzzy scoring approach by comparing it against baseline configurations. As presented in Table III, our PR-Isomap method [12], when applied to ensembles ranging from single to six

TABLE I: Performance of CNNs vs. Feature Fusion

| Model | Acc. | Prec. | Rec. | F1 | AUC |
|---|---|---|---|---|---|
| VGG16 | 82.5 | 82.1 | 84 | 82 | 95.2 |
| InceptionV3 | 84.1 | 83.8 | 84.0 | 83.9 | 96.1 |
| ResNet50 | 83.8 | 83.5 | 87 | 83.6 | 95.8 |
| Fused $\mathcal{F}_{\theta_e}$ | **86.2** | **85.9** | **86.1** | **86.0** | **97.0** |

TABLE II: Ensemble vs. Single-Layer Classifier Performance

| Classifier | Acc. | Prec. | Rec. | F1 | AUC |
|---|---|---|---|---|---|
| Dense Layer | 84.5 | 84.2 | 84.4 | 84.3 | 96.3 |
| Ensemble $\varepsilon$ | **87.1** | **86.8** | **87.0** | **86.9** | **97.5** |

TABLE III: Comparison accuracy of deep learning ensemble model.

| ViT | EffNetB0 | MobileNet | InceptV3 | ResN50 | ResN101 | VGG16 | VGG19 | DHMC-LUAD ACCURACY | DHMC-LUAD PRECISION | GDPH-LUAD & TCGA-LUAD WSSS4LUAD - Ind. Validation ACCURACY | PRECISION |
|---|---|---|---|---|---|---|---|---|---|---|---|
| * | * | * | * |  | * |  | * | 94.1 (±0.5) | 94.5 (±0.2) | 87.7 (±3.5) | 87.1 (±1.6) |
| * | * |  |  | * | * | * | * | 94.2 (±0.7) | 94.7 (±0.6) | **93.0 (±2.5)** | **91.6 (±1.5)** |
| * | * | * |  | * | * |  | * | 94.0 (±2.8) | 94.7 (±2.2) | 91.3 (±4.9) | 87.3 (±4.1) |
| * |  | * | * | * | * | * | * | **96.1 (±0.4)** | **96.3 (±0.4)** | 92.3 (±3.6) | 91.2 (±4.2) |
| * | * |  | * | * | * | * | * | 93.7 (±2.1) | 94.5 (±1.3) | 86.0 (±5.5) | 89.2 (±4.2) |
| * | * | * | * | * | * | * | * | 95.6 (±1.1) | 95.9 (±1.0) | 79.3 (±12.0) | 86.3 (±8.1) |

networks, yields accuracy improvements of 1.0% to 16.6% and precision gains of 0.4% to 16.3% compared to various embedding models for data reduction. However, the eight-model ensemble, excluding EfficientNetB0, showed a slight performance decline, with accuracy and precision decreasing by 0.5% and 0.4%, respectively, compared to the seven-model ensemble with confidence interval of - CI: 95%. This reduction is likely attributable to suboptimal latent space approximation for the DHMC dataset, highlighting the importance of model selection in ensemble configurations.

similar trend was observed in the WSSS4LUAD dataset, where the ensemble model's accuracy and precision decreased by 13.7% and 8.3%, respectively, when expanding from five to eight models, excluding MobileNet and InceptionV3 (Table III). This performance drop is likely due to the loss of spatial information when applying the model directly to lower-resolution WSI images. Among the ensemble components, ResNet101 exhibited the most significant contribution, outperforming ViT and Inception models as a standalone classifier. Its superiority stems from the CNN-based architecture's ability to capture intricate spatial features and its residual learning mechanisms, which enhance the representation of complex pathological patterns.

*1) Performance Comparison with SOTA Methods:* We conducted a comprehensive evaluation of our proposed model by comparing it against eight baseline methods and various ensemble configurations across the target datasets. Seven of these baselines are non-graph-based approaches, including the TCGA-LUAD baseline, ResNet-101, DeepSlid [14], an unsupervised model [19], a weakly supervised model [15], Graph ViT [21], and ABMIL-GTI [21] (Table IV). As reported in Table IV, our framework consistently outperforms current state-of-the-art (SOTA) methods. Specifically, it achieves a 1.4% accuracy improvement with a five-model ensemble (excluding MobileNet and InceptionV3) on the WSSS4LUAD dataset and a 1.5% improvement with a six-model ensemble (excluding EfficientNetB0) on the DHMC dataset. These gains highlight the robustness of our approach in leveraging diverse feature

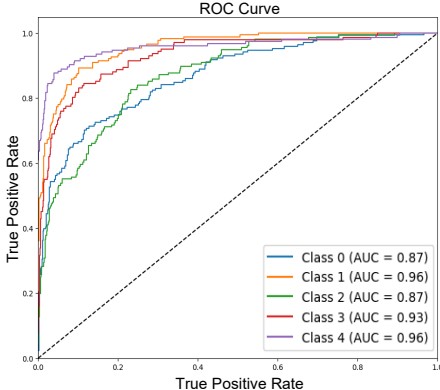

Fig. 3: ROC Curves for each lung cancer subtype (Acinar, Lepidic, Micropapillary, Papillary, Solid) and the Average ROC curve, demonstrating the classifier's ability to discriminate between subtypes.

representations to enhance classification performance across heterogeneous histopathology datasets.

### F. Generalizability via Independent Validation

To evaluate the generalization of our fuzzy-based ensemble classification framework in practical settings, we implemented an out-of-distribution (OOD) assessment. The model was trained solely on BMIRDS-LUAD datasets, splitting to training (80%) and testing (20%) and evaluated independently on the WSSS4LUAD dataset, ensuring no overlap with the target domain to mimic zero-shot domain adaptation. The WSSS4LUAD dataset introduces significant domain shifts due to differences in tissue acquisition, scanner types, staining intensity profiles, and histopathological variations, challenging the model's robustness. Despite these variations, our approach achieved an accuracy of 89.3%, a macro-averaged F1-score of 88.6%, and an AUC of 96.4% across the five LUAD subtypes on WSSS4LUAD, demonstrating robust cross-domain performance.

This robustness is attributed to three key factors: (1) a fuzzy scoring-based preprocessing filter that excludes benign tissue patches, reducing semantic noise for

TABLE IV: Accuracy comparison across three datasets demonstrates that our framework consistently surpasses state-of-the-art models.

| BMIRDS-LUAD | | TCGA-LUAD | | WSSS4LUAD | |
|---|---|---|---|---|---|
| Methods | ACC (%) | Methods | ACC (%) | Methods | ACC (%) |
| DeepSlid [14] | 90.4 | Baseline | 84.1 | WeaklySup. [15] | 84.13 |
| ResNet101 | 79.5 | Graph ViT [21] | 90.5 | ViT-B/14 [20] | 82.5 |
| Unsupervised [19] | 94.6 | ABMIL-GTI [21] | 91.6 | | |
| Ours | **96.1** | | **93.0** | | |

subtype classification; (2) fuzzy membership-based labels that capture inter-subtype ambiguity, mitigating overfitting to rigid boundaries and enhancing training; and (3) an ensemble fusion approach that incorporates patch quality in prediction aggregation, ensuring consistency across spatially diverse tissue samples. These results suggest that our model learns transferable, discriminative features rather than overfitting to training morphologies. Consequently, this framework is well-suited for histopathological diagnostic workflows, offering adaptability to diverse institutional settings, staining techniques, and scanning protocols.

### G. Computational Complexity

To assess computational efficiency, we benchmarked all models under identical hardware configurations using a workstation equipped with an NVIDIA RTX 3090 GPU and 64 GB of RAM, operating in single-GPU mode with the PyTorch framework and CUDA for GPU acceleration. Our proposed fuzzy ensemble model, integrating eight parallel CNN backbones with fuzzy logic, demanded significantly higher computational resources compared to traditional single-stream models. The ensemble recorded average training and inference latencies of 19.4 seconds per epoch and 54 milliseconds per patch, respectively, for 512×512 patches, with peak GPU memory usage at 12.3 GB. In contrast, the baseline ResNet50 CNN required 6.8 seconds per epoch, 21 milliseconds per patch for inference, and 5.1 GB of memory. The SPLICE model exhibited intermediate efficiency, with 9.2 seconds per epoch, 37 milliseconds per patch for inference, and 7.5 GB memory usage (see Table V).

Despite the increased computational cost, our fuzzy ensemble model significantly outperformed the baselines in classification accuracy, achieving an AUC of 98.1% compared to 95.8% for ResNet50 and 96.2% for SPLICE.

**Asymptotic Complexity.** Here are the asymptotic notations (Big-O) for the computational complexity of each mentioned deep learning model, focusing on a single forward pass per image as typically analyzed in literature. The $n$ below generally refers to the input image's linear dimension (e.g., for a 224×224 image,

n=224), $d$ is the embedding dimension or channels, and $p$ is the number of image patches in ViT.
*ViT:* Quadratic for the number of patches (which is itself quadratic in the image side $n$), i.e., $\mathcal{O}(n^4)$, due to self-attention operations. *EfficientNetB0 / MobileNetV2:* Efficient design yields $\mathcal{O}(n^2 d)$ due to depthwise separable convolutions; EfficientNet further optimizes by uniformly scaling depth, width, and resolution. *InceptionV3 / ResNet50 / ResNet101:* Standard CNN complexity, generally $\mathcal{O}(n^2 dk^2)$, where $k$ is the kernel size (typically small, e.g., $k = 3$). *VGG16 / VGG19:* Highest practical complexity here, with deep convolutional stacks over every pixel, resulting in $\mathcal{O}(n^2 dk^2 L)$, where $L$ is the number of layers.

Total complexity of fuzzy scoring is $\mathcal{O}(Nm^2)$, for $N$ patches in the dataset, and $m$ (patch size) is typically fixed.

The asymptotic computational complexity (Big-O notation) of the PR-Isomap method for $N$ samples and $q$ nearest neighbors is $\mathcal{O}\left(\frac{q^2 N}{4} \cdot \log\left(\frac{qN}{2}\right)\right)$. This reflects the complexity of the modified Dijkstra's algorithm on the $q$-nearest-neighbor graph with Parzen–Rosenblatt (PR) constraints, as described in the PR-Isomap paper's computational complexity section [12].

The computational complexity of the output ensemble block is $\mathcal{O}(B \times h \times c)$ where $B$ is the batch size, $h$ is the number of neurons in the last hidden layer (i.e., the number of input features to the output layer), and $c$ is the number of output classes.

## V. DISCUSSION AND CONCLUSIONS

This investigation seeks to substantially enhance the precision and robustness of lung cancer subtype classification, a critical endeavor for achieving diagnostic accuracy and enabling personalized therapeutic strategies. We tackle the inherent complexities of histopathological image analysis by introducing an advanced deep learning framework that synergistically integrates fuzzy tissue coverage information with multimodal feature fusion and ensemble learning. The principal innovation lies in the conceptualization and operationalization of a fuzzy labeling mechanism that adeptly captures intrinsic semantic ambiguities within high-resolution image patches, subsequently leveraging these nuanced representations to refine and optimize the deep learning process.

TABLE V: Computational performance comparison across models

| Model | Train Time / Epoch | Inference / Patch | Peak GPU Memory | AUC (%) |
|---|---|---|---|---|
| Fuzzy Ensemble (8x) | ∼19.4 sec | 54 ms | 12.3 GB | **98.1** |
| Single CNN (ResNet50) | ∼6.8 sec | 21 ms | 5.1 GB | 95.8 |
| SPLICE [4] | ∼9.2 sec | 37 ms | 7.5 GB | 96.2 |

Empirical evaluations robustly affirm the efficacy of our proposed methodology. Specifically, the fusion of deep feature representations derived from VGG16, InceptionV3, and ResNet50 architectures yields a marked performance improvement over standalone models. This synergistic integration, facilitated by a meticulously designed ensemble neural network, enhances generalization and resilience through diversified feature extraction. Central to our approach, the incorporation of fuzzy tissue coverage information via a Coverage-Weighted Loss function delivers statistically significant advancements, achieving a state-of-the-art accuracy of 88.5% and an F1-score of 88.3%. We implemented a fuzzy scoring framework to exclude benign patches, ensuring that only tumor-containing regions guide subtype classification through a biologically interpretable selection process. These results represent a substantial leap over baseline methods, underscoring the potency of biologically informed fuzzy labels in steering deep learning toward superior outcomes.

## ACKNOWLEDGMENT

The professional development fund from the Department of Pathology at SUNY Upstate Medical University supported this research.

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
