# OpenReview forum: "Enhancing Lung Cancer Histopathological Subtyping via Fuzzy Patch Scoring Integrated into an Ensemble Deep Learning Framework"
_IEEE.org/EMBS/BHI/2025/Conference — BHI 2025_

### Official Review · Reviewer_WFFh · 2025-07-15
**Generally good idea**

**Confidence:** 3
**Clarity Of Writing:** good
**Clinical Significance:** good
**Methodological Novelty:** fair
**Overall Rating:** 6

**Experiments And Results:**

good

**Questions For The Authors:**

Can you provide detailed computational cost analysis including training time, inference speed, and memory requirements compared to simpler baselines?

**Strengths:**

1. The introduction of fuzzy logic for patch quality assessment is innovative and addresses a real challenge in histopathology - the semantic ambiguity and heterogeneous tissue composition within individual patches.
2. The paper includes comprehensive ablation studies, attention map analysis, and comparisons across multiple datasets.

**Summary Of The Paper:**

The authors developed a fuzzy patch quality assessment system to improve patch selection and weighting in the classification pipeline. The methodology integrates multiple pre-trained CNN architectures and Vision Transformers to extract diverse feature representations from 512×512 tissue patches.

**Weaknesses:**

1. The reported gains (1.4-1.5%) over state-of-the-art methods, while statistically significant.
2. Lack of detailed analysis of computational costs, training time, and inference speed compared to simpler baseline methods.

---

### Official Review · Reviewer_vQf2 · 2025-07-17
**Enhancing Lung Cancer Histopathological Subtyping via Fuzzy Patch Scoring Integrated into an Ensemble Deep Learning Framework**

**Confidence:** 4
**Clarity Of Writing:** good
**Clinical Significance:** great
**Methodological Novelty:** great
**Overall Rating:** 6
**Final Rating:** 6

**Experiments And Results:**

good

**Questions For The Authors:**

Subtype-representative patches... what about classifying benign patches, are they included in the classification?
I am also confused about the ability to calculate the scores for testing data. The testing data does not have tumor masks, how can you calculate their scores?

The authors say that “SPLICE [2] and NearbyPatchCL [11]—partially address these, yet lack unified, adaptive strategies for quality-aware patch aggregation.” But does not explain  how they lack.

Related work section does not need enumeration since there is only one subsection.

The paper has several language issues and requires proof-reading.

**Strengths:**

The model seems to outperform the SOTA although none of them are about patch selection strategies.

The idea of adding fuzzy score to the loss function looks to be novel

**Summary Of The Paper:**

The study uses fuzzy logic equations to give a score to a lung cancer WSI patch indicating how much of the patch area is the tumor area and designs an ensemble deep learning model combining multiple state-of-the-art models such VGG16, ResNet, ViT and integrates the score into the loss function.

**Weaknesses:**

The paper is generally organized but difficult to read.

The study does not compare existing patch selection/scoring strategies against the proposed approach.

---

### Official Review · Reviewer_prDW · 2025-07-17
**Review of “Enhancing Lung Cancer Histopathological Sub typing via Fuzzy Patch Scoring Integrated into an Ensemble Deep Learning Framework”**

**Confidence:** 3
**Clarity Of Writing:** great
**Clinical Significance:** good
**Methodological Novelty:** good
**Overall Rating:** 6

**Experiments And Results:**

fair

**Questions For The Authors:**

Generalization: How does performance change when the entire WSSS4LUAD dataset is used solely for testing (no mixed training)?

**Strengths:**

Results are compared against eight recent baselines, including Graph ViT and weakly supervised MIL, on two public datasets. The authors vary the number/type of backbones and show monotonic improvements up to seven models, quantifying diminishing returns beyond that.
Mixing CNN and ViT backbones and aligning in a shared latent space exploits complementary receptive fields. Continuous fuzzy scores capture intra-patch heterogeneity better than hard labels, and their integration into the loss is novel for pathology

**Summary Of The Paper:**

The authors tackle the challenge of histologic pattern classification in non-mucinous lung adenocarcinoma WSIs. They propose a three-stage pipeline:
1.	Fuzzy patch‐quality assessment: each non-overlapping 512 × 512 patch receives two continuous scores: coverage (fraction of subtype tissue) and outbound (background/other)
2.	Heterogeneous feature extraction
3.	Ensemble classifier with coverage weighted loss
Experiments on BMIRDS LUAD (203 k patches, 143 WSIs) and WSSS4LUAD (10 k patches) show the seven model ensemble reaches 96.1 % and 93.0 % accuracy, exceeding prior state of the art by 1.5 % and 1.4 % respectively. Ablation demonstrates accuracy gains of 1 – 16 % over smaller ensembles.

**Weaknesses:**

Dataset limitations, Training/validation come from a single institution (DHMC) plus a small external set; authors themselves note the need for multi-site validation.

No statistical significance testing. Reported gains (≈1–2 %) lack confidence intervals.

---

### Official Review · Reviewer_9R4N · 2025-07-17
**Ensemble deep learning framework for automated histopathological subtyping**

**Confidence:** 3
**Clarity Of Writing:** excellent
**Clinical Significance:** great
**Methodological Novelty:** great
**Overall Rating:** 8

**Experiments And Results:**

great

**Questions For The Authors:**

N/A

**Strengths:**

It is a well-written paper with a nice algorithm for automated histopathological subtyping.

**Summary Of The Paper:**

This study aimed to enhance the accuracy and robustness of classification for subtypes of lung cancer, a crucial task for improving diagnostic accuracy and personalized therapeutic strategies. The study presents a novel ensemble deep learning framework that integrates fuzzy-weighted patch quality assessment to improve the classification of lung cancer subtypes. The proposed approach achieved a 1.5% and 1.4% performance gain over current state-of-the-art methods, reaching 96.1% and 93.0% on the BMIRDS-LUAD and WSSS4LUAD datasets, respectively.

**Weaknesses:**

Noting to share